# How to Evaluate Fibrosis in IBD?

**DOI:** 10.3390/diagnostics13132188

**Published:** 2023-06-27

**Authors:** Helena Tavares de Sousa, Fernando Magro

**Affiliations:** 1Gastroenterology Department, Algarve University Hospital Center, 8500-338 Portimão, Portugal; 2ABC—Algarve Biomedical Center, University of Algarve, 8005-139 Faro, Portugal; 3Unit of Pharmacology and Therapeutics, Department of Biomedicine, Faculty of Medicine, University of Porto, 4200-450 Porto, Portugal; fm@med.up.pt; 4Department of Gastroenterology, São João University Hospital Center, 4200-319 Porto, Portugal; 5CINTESIS@RISE, Faculty of Medicine, University of Porto, 4200-450 Porto, Portugal

**Keywords:** inflammatory bowel disease, fibrosis, inflammation, imaging, biomarkers

## Abstract

In this review, we will describe the importance of fibrosis in inflammatory bowel disease (IBD) by discussing its distinct impact on Crohn’s disease (CD) and ulcerative colitis (UC) through their translation to histopathology. We will address the existing knowledge on the correlation between inflammation and fibrosis and the still not fully explained inflammation-independent fibrogenesis. Finally, we will compile and discuss the recent advances in the noninvasive assessment of intestinal fibrosis, including imaging and biomarkers. Based on the available data, none of the available cross-sectional imaging (CSI) techniques has proved to be capable of measuring CD fibrosis accurately, with MRE showing the most promising performance along with elastography. Very recent research with radiomics showed encouraging results, but further validation with reliable radiomic biomarkers is warranted. Despite the interesting results with micro-RNAs, further advances on the topic of fibrosis biomarkers depend on the development of robust clinical trials based on solid and validated endpoints. We conclude that it seems very likely that radiomics and AI will participate in the future non-invasive fibrosis assessment by CSI techniques in IBD. However, as of today, surgical pathology remains the gold standard for the diagnosis and quantification of intestinal fibrosis in IBD.

## 1. The Importance of Fibrosis in IBD

Inflammatory bowel disease (IBD), which includes Crohn’s disease (CD) and ulcerative colitis (UC), is characterized by chronic intestinal inflammation mediated by dysregulated immune responses to such factors as diet and microbiota [1,2,3,4].

In both CD and UC, chronic inflammation causes disruption of the epithelial barrier and tissue destruction. Fibrosis, which is a healing mechanism, becomes progressive and damaging in the scope of long-lasting IBD, in which persistent tissue damage and healing result in scar tissue formation [5,6]. At tissue and cellular levels, fibrosis is an amplified response characterized by the accumulation of collagen-rich extracellular matrix (ECM) produced by an increased number of mesenchymal cells, including fibroblasts, myofibroblasts, and smooth muscle cells (SMCs) [7]. The proliferation of fibroblastic cells, along with the accumulation of ECM, are the hallmarks of intestinal strictures in IBD [8]. Fibrosis is a frequent outcome in the natural history of IBD and is the background for most of the IBD complications, such as strictures, bowel penetration, and obstruction, often demanding surgery [5,6,9]. It has been estimated that about 30% to 50% of CD patients and 1% to 12% of UC patients would suffer from fibrosis complications during the disease course [6,10,11]. Until recently, intestinal fibrosis was considered an unavoidable complication of IBD in patients that did not respond to anti-inflammatory therapy, often requiring surgical intervention [6]. The emergence of the possibility of an anti-fibrotic approach changed this paradigm, creating challenges in terms of diagnosis and treatment of bowel fibrosis [6,12]. As such, understanding the molecular and cellular mechanisms underpinning fibrosis and improving techniques for the assessment of fibrosis in IBD patients are still relevant research topics.

### 1.1. Fibrosis in CD

In CD, both inflammation and fibromuscular changes are transmural, leading to progressive thickening of the bowel wall and stricture development, even in the absence of inflammation. Pathologically, intestinal fibrosis in CD is characterized by ECM accumulation and mesenchymal cell expansion affecting all layers of the bowel wall along the intestinal tract [13]. In addition, recent pathologic consensus defined small bowel strictures in CD as a combination of decreased luminal diameter and increased thickness of all layers of the intestinal wall, including expansion of the muscularis mucosae (MM) and inner muscularis propria (MP), muscularization of the submucosa, and fibrosis of the submucosa and intestinal wall [14]. Notwithstanding, the universality of this concept was recently challenged by the description of a non-hypertrophic, constrictive type of stricture in CD [15]. Regardless of the type of stricture, these remain common complications of CD with serious clinical relevance and impact on the patients’ quality of life [12,15,16].

Aside from fibrosis in strictures, it has been proposed that a certain degree of fibrosis would exist in nearly all CD phenotypes, even from early onset. In addition, it has been demonstrated that the degree of fibrosis may be similar in both stricturing and penetrating CD, with differences regarding the degree of transmural inflammation [17].

Though still used in clinical practice, the classification of CD in three-category phenotypes, as inflammatory or non-stricturing, non-penetrating (B1), stricturing (B2), and penetrating (B3) disease, is now considered too rigid [18,19]. As an alternative, CD shall be viewed through the lens of a progressive accumulation of intestinal fibrosis and damage over the course of the disease, leading to stricturing and/or penetrating complications, as supported by epidemiological natural history studies [16,20,21,22,23,24,25,26]. This progressive and cumulative structural bowel damage would occur irrespective of symptoms and, considering current fibrogenesis knowledge, of the degree of intestinal inflammation [6,27,28]. Hence, clinical symptoms, disease activity [29,30], and progression of bowel damage [4,31] are not totally correlated.

Considering that population-based studies have shown a 10-year cumulative risk of surgery between 40% and 71% [32,33,34] and that fibrosis is a marker of advanced disease, its importance is central in the setting of CD, as it underlies the need for surgical resection in stricturing disease and, maybe, also in penetrating complications, as strictures coexist in over 85% of penetrating CD [4,30,31,32,33,34,35].

In this context, to further understand pathology changes in CD, deep research on the basic cellular and molecular mechanisms of fibrogenesis is warranted.

### 1.2. Fibrosis in UC

In UC patients, fibrosis is characterized by a thickening of MM and excessive ECM deposition in the submucosa, affecting deeper layers only after profound ulceration of the submucosa [36,37,38]. Strictures are uncommon in UC, and the majority are benign and reversible [39]. However, in UC, fibrosis originates the increased wall stiffness, which may result in motility abnormalities, anorectal dysfunction, rectal urgency, and incontinence [10,38].

The evidence regarding fibrosis in UC is limited and controversial, but a comprehensive assessment performed by Gordon et al. in 2018 demonstrated that UC is characterized by progressive fibrosis and MM thickening in correlation with the severity and chronicity of inflammation. Hence, deep remission, including histological remission, should be a priority and a therapeutic target [40]. Recent research in mice with dextran sulfate sodium (DSS)-induced colitis has shown that, in UC, changes in motility may also be related to neuronal modification. The study highlighted that UC does not promote neuron death but induces changes in the chemical code of myenteric neurons [41]. A better comprehension of these data and the translation of these results depend on studies on human tissue.

## 2. Is There Fibrosis without Inflammation?

During physiological tissue repair, collagen substitutes the temporary fibrin plug to create a permanent sealant of the injured tissue. Briefly, in response to injury signals, fibroblasts activate, proliferate, expand, and transform into myofibroblasts, which have the innate contractile ability and produce higher levels of ECM components. Under normal circumstances, once the healing process is accomplished, the fibrotic matrix is degraded by matrix metalloproteases (MMPs), and both fibroblasts and myofibroblasts undergo apoptosis or revert to a non-activated state [42,43]. However, in the setting of recurrent or persistent epithelial injury, intestinal inflammation initiates and sustains fibrogenesis, which can progress even after the inflammatory trigger has subsided [6]. Chronic epithelial and endothelial injury release chemotactic factors that promote recruitment and constitutional activation of immune and mesenchymal cells, leading to inflammation-dependent and -independent progression of fibrosis with progressive organ dysfunction [43].

### 2.1. Inflammation-Dependent Fibrogenesis

In CD patients, research has highlighted a strong connection between inflammation and fibrosis (Figure 1). In fact, in intestinal resection specimens, both components generally overlap and share a similar distribution [6,44,45,46].

Inflammation has been established as the most potent activator of mesenchymal cells, initiating fibrogenesis, both in the early stages of CD and over the course of the disease, eventually leading to a fibrotic scar that may permeate the whole tissue architecture [47].

The physiopathological process of inflammation-dependent fibrogenesis is complex and involves a variety of such molecules and cells as immune cells, ECM-producing cells, and intestinal microbiota [48]. Without exhausting all involved mechanisms, it is known that Th17 and Th2 cells play a central role in IBD inflammation and fibrosis through the secretion of interleukins (ILs) involved in intestinal myofibroblasts’ activation, migration, and ECM production (IL17, IL21) [11]; epithelial to mesenchymal transition (EMT, IL17) [11,45]; collagen deposition by fibroblasts (IL13); secretion of latent transforming growth factor (TGF)-β and MMP-9 by macrophages; and TGF-β activation by cleaving its latency-associated peptide (LAP) [11,43,45,48]. TGF-β is considered the major cytokine in intestinal fibrosis [43,48,49], mediating the differentiation of fibroblasts into myofibroblasts and promoting myofibroblasts’ proliferation, migration, contraction, and resistance to apoptosis, while increasing the production of ECM components and tissue inhibitor of MMP (TIMP)-1 [11,50,51]. Importantly, TGF-β has an anti-inflammatory role, including the promotion of class-switching immunoglobulin A, inhibition of antibody production, and downregulation of inflammatory cytokine production by monocytes and macrophages through inhibition of nuclear factor (NF)-B. Overall, TGF-β is key in keeping the immune balance of the intestine through enhancing mucosal defense and tissue healing, promoting immune tolerance, and suppressing anti-inflammatory responses [52]. Apart from TGF-β, many other soluble factors produced by immune cells during inflammatory responses can promote fibrogenesis, such as transforming necrosis factor (TNF), IL-23, IL-36, activins, connective tissue growth factor (CTGF), epidermal growth factor (EGF), insulin-like growth factor (IGF)-1 and -2, platelet-derived growth factor (PDGF), vascular growth factor (VGF), galectin-3, endothelins (ET-1, -2 and -3), products of oxidative stress, components of the renin-angiotensin system (RAS), and mammalian target of rapamycin (mTOR) [6,11,43,45,48,49,50,51].

After T cells, macrophages are also expanded in IBD, although their specific role in intestinal fibrosis is only partly explained [45]. It has been described that IL-36α secreted from M1 macrophages locally induces myofibroblasts proliferation and collagen VI production [53], while an increase in M2 macrophages was demonstrated in creeping fat [54]. However, it was shown that, in certain situations, M2 cells could inhibit ECM synthesis; thus, their definitive role in fibrosis requires further investigation [48]. Neutrophils, mast cells, and eosinophils can all promote fibrosis through the release of pro-fibrotic cytokines, chemokines, and, for neutrophils, reactive oxygen and nitrogen species, while basophils have a less clear role in fibrogenesis [43].

Considering that the hallmark of CD strictures includes excessive secretion of ECM and increased numbers of mesenchymal cells in distinct locations of the bowel wall, the importance of ECM-producing cells, namely, fibroblasts, myofibroblasts, and SMCs in fibrosis is crucial [6]. Histopathologically, these cells can be distinguished according to the expression of vimentin (V), α-smooth muscle actin (α-SMA), and desmin (D): fibroblasts exhibit a V(+), α-SMA (−), D (−) pattern, myofibroblasts a V(+), α-SMA (+), D (−) one, and SMCs are V(− or low), α-SMA (+), D (+) [45]. Fibroblasts are the primary effector cell in CD fibrosis. As referred, several mediators (the strongest being IL13 and TGF-β) drive the fibrotic response of intestinal fibroblasts in IBD [11,48,55]. Once activated, fibroblasts proliferate, migrate, increase the secretion of ECM, and can transform into myofibroblasts [56]. The same occurs with activated myofibroblasts, which can produce much higher levels of ECM than intestinal fibroblasts [45,57]. Moreover, if dysregulated, activated myofibroblasts may shift collagen type IV synthesis to collagens type I and III, which will be gradually deposited into fibrillar ECM, distort normal architecture, and increase tissue stiffness and scarring [58]. In an inflammatory environment, they can differentiate from fibroblasts and SMCs, but also from other cell types, such as fibrocytes, pericytes (blood vessels walls’ fibroblasts), and epithelial, endothelial, stellate, or bone-marrow-derived stem cells [6,11,43,47,48,58]. Moreover, under chronic inflammation, SMCs can transdifferentiate into myofibroblasts and vice-versa [43]. Once activated by TGF-β1 and IL-1β, SMCs increase IL-6 production, thus further contributing to intestinal inflammation [59]. Apart from originating myofibroblasts, contributing directly to fibrogenesis, SMCs are the major contributors to intestinal wall thickening in CD strictures, both by hyperplasia and/or hypertrophy of muscular layers and by undergoing fibromuscular hyperplasia in the submucosa [47,60,61,62]. To finalize, a reference to creeping fat is mandatory. Creeping fat, defined as a pathologically increased fat tissue adjacent to areas of the intestine affected by CD, has been shown to correlate both to strictures and degree of inflammation [63] and, more recently, to MP hyperplasia through free fatty acids-mediated intestinal SMCs hyperplasia [8,64]. Conversely, Ren Mao et al. demonstrated that activated MP SMCs interact with creeping fat preadipocytes through the production of a specific ECM scaffold able to induce preadipocytes migration out of mesenteric fat into de novo creeping fat [64]. Creeping fat directly promotes inflammation-dependent fibrosis in the adjacent intestine through large amounts of pro-fibrotic cytokines, adipokines, growth factors, and fatty acids produced by both innate and adaptive immune cells as well as adipocytes [65]. On the other hand, creeping fat fibrosis is a well-known histopathological feature of CD, which was described by Karl Geboes as “fibrous strands are present in the mesenteric fat, irradiating from the intestine and surrounding thickened, hypertrophied fat lobules” [66]. The mechanisms underlying the concept of penetrating fibrosis and creeping fat fibrosis are complex and not fully understood, involving fibroblasts, SMCs, preadipocytes, and macrophages (specifically M2-subtype [67,68]. Serum markers for microbiota were also associated with complicated and stricturing CD [69]. In addition, intestinal dysbiosis and its secondary products have been shown to be able to induce fibrosis in the gut of CD patients [70]. However, it is not clear if dysbiosis affects only inflammation-dependent fibrogenesis or also the -independent one, and, if so, which microbiota components promote fibrosis without inflammation [48].

### 2.2. Inflammation-Independent Fibrogenesis

Despite our increasing capacity to control intestinal inflammation through such drugs as biologics and new small-molecule drugs, the progress in preventing progression to fibrosis and stricture development is minimal [43]. Moreover, while suppressing inflammation waives inflammatory markers, it does not reduce the expression of profibrotic mediators, suggesting the existence of inflammation-independent mechanisms mediating self-perpetuating fibrogenesis [6].

In the absence of inflammatory stimuli, ECM stiffness and mechanotransduction by fibroblasts (Figure 2) should be considered the central inflammation-independent mechanisms of intestinal fibrosis [71]. In fact, even though ECM is accountable for keeping tissue integrity, it is a dynamic structure able to communicate with a variety of cells, including those involved in the production of its own constituents. The interaction of ECM with fibroblasts includes multi-protein assemblies at the cell membrane, called focal adhesions (or focal adhesion complexes). ECM stiffness is determined by the abundance of fibrillary collagens and their degree of cross-linking, as well as the degree of hydration of the matrix, determined by the concentration of proteoglycans and hyaluronic acid and also the coexistence of inflammatory edema [71]. As referred, through the course of IBD, cytokines and fibrotic growth factors mediate the deposition and crosslinking of ECM components, making ECM stiffer with changes in its mechanical properties. These changes appear to activate pro-fibrotic signaling cascades in fibroblasts that only recently began to be explored. Briefly, ECM stiffness perpetuates fibrogenesis through the activation of mesenchymal cells, which, in turn, can further increase stiffness and regulate contraction in an inflammation-independent way [6,72]. It has been shown that in the absence of inflammation, tissue stiffness alone can lead to the progression of fibrosis in CD by inducing a morphological transformation of intestinal myofibroblasts from round to stellate shape, cellular proliferation, collagen and αSMA production and development of focal adhesions [73]. The mechanical properties of ECM stiffness are able to induce profibrotic signaling cascades in fibroblasts at least by two concurrent mechanisms [71]. First, the increased stiffness of the regenerated ECM drives fibroblast differentiation through the focal adhesion-mediated translation of ECM mechanical forces to biochemical activity within the cell, mediated by a dynamic cellular cytoskeleton. Cells adhered to ECM respond to resistance changes by increasing both the number and size of focal adhesions and the cytoskeletal pre-stress, increasing F-actin/myosin stress fibers and downstream intracellular pro-fibrotic signaling to increase ECM deposition. Second, through mechanotransduction by fibroblasts—a process by which fibroblasts convert mechanical signals into biochemical signals—ECM stiffness can also lead to the release of the potent pro-fibrotic and anti-apoptotic TGF-β “stored” in the ECM, creating a positive feedback loop crucial for sustained myofibroblast function [71,74].

The process is complex and involves a variety of structures and molecules; integrins have been demonstrated to play an important role as components of focal adhesion complexes [71]. The most important integrins involved in mechanotransduction are α5β1, which is expressed in fibroblasts [73], and αγ class integrins [71]. Herein, it is worthwhile to recall the role α5β1integrin in mediating the ability of fibronectin of the SMCs-derived matrisome adjacent to the outer aspect of the MP to induce preadipocytes migration out of mesenteric fat leading to de novo formation of creeping fat. Moreover, no proinflammatory cytokine could promote this migration [64], reinforcing the importance of this molecule in inflammation-independent fibrogenesis in CD.

In the ECM, TGF-β is kept inactive through its inclusion in the so-called TGF-β-large latent complex, composed of a latency-associated peptide (LAP) bounded to a latent TGF-β binding protein (LTBP), which, in turn, is linked to collagen and proteoglycans (as decorin, thrombospondin, and fibronectin), determining its bioavailability [53,75].

In resting conditions, fibroblasts form only loose contacts with ECM through weak and short-lived integrin adhesions. Upon tissue injury, fibroblasts rapidly transform into an activated state, secreting mainly fibronectin and fibronectin ED-A and migrating over the injured tissue to restore tissue integrity. Only in the late contracting phase of wound healing, by means of a combination of mechanical stimulations through focal adhesions and TGF-β presence, will fibroblasts terminate their differentiation into myofibroblasts by expressing αSMA [51,76]. Importantly, after wound healing, the contracting ECM is always less organized and more rigid than the original one [6,71].

Hence, the ECM functions as a reservoir of both pro-fibrotic and pro-inflammatory mediators, ready to be released upon mechanical stretching of the stiff ECM, leading to fibrosis either directly (without inflammation drivers) or indirectly through the initiation of inflammatory cascades [71]. How cells, namely, fibroblasts and myofibroblasts, integrate the mechanical and biochemical information present in the ECM to impact cellular functions is still not understood.

### 2.3. Unmet Needs

Considering that inflammation is the most potent trigger for fibrogenesis in CD, the early control of inflammation should reduce the incidence of strictures in the long term. Although this was not proven before the biologic era [68,69], it has been shown that early biologic therapy may achieve this purpose [77,78,79], leading to a reduction in hospitalizations and surgery [80]. However, as stated before, current anti-inflammatory and immunosuppressive strategies have not yet been capable of fully controlling tissue remodeling and fibrosis progression nor eliminating established complications.

In this setting, preventing intestinal fibrogenesis or reversing already established bowel strictures in patients with IBD should remain the ultimate goal of disease management [35]. For the end purpose of developing clinical trials for anti-fibrotic agents that would change the IBD treatment paradigm, the Stenosis Therapy, and Anti-fibrosis Research (STAR) consortium has been dedicated to the revision of the available knowledge on the identification and characterization of strictures in CD, in which fibrosis is a major issue [81,82,83]. Based on the collected data, the STAR consortium has been working on definite standards to measure response to anti-fibrotic agents by developing endpoints and standard methodology for clinical, radiological, and histopathologic scoring systems essential for the design of reliable anti-fibrotic clinical trials [83,84,85,86].

## 3. Non-Invasive Techniques to Access Fibrosis

Several clinical, genetic, and serological risk factors for complicated or disabling disease and/or surgery have been identified in the setting of IBD. Among these, only clinical factors are being used in clinical practice to select patients who would benefit from early medical aggressive therapy [31]. However, none of these risk factors has been undoubtedly associated with stricturing disease or proved to predict fibrosis development [11,87].

Considering that, at this point, fibrosis cannot be predicted, IBD management would benefit from an accurate and non-invasive assessment of intestinal fibrosis, as patients with predominant inflammatory strictures are likely to respond better to current therapies, whereas those with established fibrotic ones will probably require surgery [6,88,89]. However, as stated, pure fibrotic or inflammatory strictures in surgical specimens are rare, with inflammation and fibrosis usually co-existing in varying degrees [6].

Broadly, fibrosis assessment in IBD is a challenging and non-invasive evaluation of fibrosis remains elusive. Despite the advances in imaging and molecular technologies, definitive identification, characterization, and quantification of intestinal fibrosis in CD still depend on the histopathological evaluation of surgical specimens [87], even though no histopathological scoring system has been validated or widely accepted for this purpose [14,82,85,90,91,92]. To some extent, the same applies to UC, in which fibrosis affects mostly MM but can sometimes extend to the submucosa. Thus, its complete quantification also cannot rely solely on endoscopic biopsies or imaging [10,38].

While some imaging techniques have been intensively investigated through the last decade, mainly ultrasound elastography [93,94,95] and specific magnetic resonance techniques, such as diffusion-weighted imaging (DWI) [96,97,98,99,100,101], diffusion kurtosis imaging (DKI) [102,103], magnetization transfer (MT) [104,105,106,107,108,109], and intravoxel incoherent motion imaging (IVIM) [110,111], none has been definitively established as reliable for this objective [112,113]. Very recently, some promising developments arose through artificial intelligence (AI) techniques in cross-sectional imaging, such as radiomics [114,115,116,117]. In addition, biomarkers remain a field of intense investigation and deserve proper discussion [42,83,118,119].

### Imaging Techniques

Even though imaging techniques have been used for several decades to identify and measure the severity of IBD, only in recent years has research explored methods with valuable prognostic value [120].

#### Cross-Sectional Imaging

Among imaging techniques, cross-sectional imaging (CSI) is an essential tool for IBD characterization in all disease stages [81,106,121,122,123,124]. Traditionally, CSI has been used to evaluate the extent and activity of CD and to detect such complications as abscesses or fistulae, but it is also being used for the assessment of treatment response and prediction of outcomes and post-surgery recurrence [106,121].

It is recommended that such CSI techniques as ultrasonography (US), computed tomography (CT), enterography (CTE), or magnetic resonance (MR) enterography (MRE) should be performed at the time of diagnosis of CD to complement endoscopy by assessing stricturing and penetrating complications [110,125,126,127,128,129]. These techniques are all able to detect strictures with high accuracy, with the selection of the best approach depending on availability, cost, patient clinical status (including comorbidities), and radiation concerns [124]. Their ability to identify and quantify fibrosis has been variably studied mostly in CD (Table 1).

##### Magnetic Resonance

MRE is considered the most advanced technique for imaging fibrosis in CD strictures. Its accuracy and lack of radiation exposure are the most attractive features. Overall, the sensitivity of MRE for stricture detection ranges from 75% to 100%, with an estimated specificity between 91% and 96% [81,124,130,131,132]. As for fibrosis assessment, MRE has been described to accurately differentiate between severe and mild–moderate fibrosis, with a sensitivity of 0.94 and a specificity of 0.89 (*p* < 0.01), irrespective of the degree of inflammation [133]. In addition, MRE could also distinguish severe fibrosis from severe muscle hypertrophy in ileal CD [134]. Recently, an MRE-based composite score was shown to be a very good predictor of histologic fibrosis (ROC_AUC_ = 0.910) [128].

Concerns related to the intravenous administration of gadolinium justify the efforts to replace MRE with other MR techniques that do not demand intravenous contrast [106]. Even though diffusion-weighted imaging (DWI) has been used to detect inflammatory activity in CD [97], its utility for fibrosis assessment based on the assumption that the presence of fibrotic tissue is related to the restricted diffusion of water molecules, is still not defined [81,106,124,135]. Initial research showed that fibrosis was associated with low attenuated diffusion coefficient (ADC) values, presumably due to the reduction in extracellular space in fibrotic tissue leading to a restriction in diffusion [96,98,99,136]. However, more recent data evidenced constraints while distinguishing severe fibrosis from severe muscle hypertrophy in ileal CD [134], while others showed that the accuracy of DWI in detecting fibrosis varies with the degree of bowel inflammation. Since the available reports on the use of DWI on IBD included a wide range of ADC values, threshold values have not yet been defined to differentiate between active inflammatory, non-active, and fibrotic disease [98,136,137,138]. Mainenti et al. associated this variability with technical aspects, such as differences in MR equipment concerning magnetic field strengths, lack of reproducibility, and absence of standardized sequence parameters [100,101]. Still, even though DWI is not validated as a reliable quantitative biomarker for fibrosis, its short analysis time, absence of contrast, ability to provide qualitative and quantitative data, and high accuracy for inflammation and penetrating complications in IBD support the continuous research on its utility in the setting of fibrosis assessment in IBD.

However, researchers have highlighted relevant inconsistencies in DWI, such as the concept that, in this method, ADC calculation assumes that water distribution obeys a Gaussian model, not reflecting the impact of cell structures and biophysical properties on water displacement [139]. In response to this inconsistency, MR scanners evolved to consider non-Gaussian diffusion, demanding distinct analysis models. In this setting, diffusion kurtosis imaging (DKI) emerged as a more robust analysis model, providing a more precise display of water diffusion in the human body than conventional DWI [139,140]. DKI was first applied in the context of IBD to evaluate CD activity, providing values of K_app_ (apparent diffusional kurtosis) and D_app_ (diffusional coefficient) corrected for non-Gaussian behavior, which could distinguish between inactive, mild, and moderate–severe CD (*p* < 0.05) with better accuracy than DWI [139]. Concerning fibrosis, DKI has been considered useful for staging liver fibrosis in a rabbit model [141]. In the scope of IBD, it has been shown that K_app_ was significantly correlated to fibrosis grades and allowed to distinguish between the absence of fibrosis or mild fibrosis and moderate to severe fibrosis (sensitivity of 95.9% and specificity of 78.1%), evidencing its potential for the assessment of bowel fibrosis [102]. However, further studies are warranted to validate this data.

Based on previous data on animal models [105,108], magnetization transfer–magnetic resonance (MT-MR) was explored in a cohort of 31 CD patients. The results showed that magnetization transfer ratio (MTR) values correlated with fibrosis (*p* < 0.0001) [96], confirming that MT-MR may be of value for fibrosis identification in CD with only a small increase in the analysis time. In 2018, Li et al. confirmed that MTR was strongly correlated with fibrosis scores (r = 0.769, *p* = 0.000) but not with inflammation scores (r = −0.034, *p* = 0.740) and could differentiate moderately–severely fibrotic from non-fibrotic and mildly fibrotic bowel walls. This study showed its superiority when compared to DWI-MR and contrast-enhanced MR [107].

In recent years, the concept of textural analysis (TA) was introduced in MR imaging, namely, for fibrosis detection purposes. In this line, MR elastography was presented by Avila F and colleagues in a pilot study in which the tissular stiffness value, measured by MR elastography, correlated with the degree of fibrosis (*p* < 0.001), according to an MR-based score [142]. No pathological correlation was undertaken; hence, the true value of this technique remains unproven. Very recently, TA of T2-weighted MR imaging (T2WI) was used to assess intestinal fibrosis in a dextran sodium sulfate (DSS) murine model and, as a proof-of-concept in 5 CD patients, against MT-MR and histopathology. TA features included skewness, kurtosis, and entropy. Both entropy and MT ratio correlated with histopathological fibrosis (r = 0.85 and r = 0.81, respectively); MT was superior in monitoring bowel fibrosis when coexisting with inflammation (linear regression R^2^ = 0.93 vs. R^2^ = 0.01, respectively) as well as in assessing antifibrotic response in mice. As entropy increased with fibrosis accumulation in human CD strictures, TA was capable of quantifying fibrosis in mixed inflammatory–fibrotic strictures. Considering that TA of T2WI is an accessible post-processing technique, the authors conclude that it deserves further research and validation both for clinical practice and antifibrotic trial design [143].

##### Computer Tomography

CTE has proved to be adequate, in terms of sensitivity and specificity, for the detection of features suggestive of CD and its complications [106,121,127]. In addition, it is widely available, fast, has relatively low cost, and enables the analysis of longer portions of the gastrointestinal tract than MR and the detection of extraintestinal manifestations [124,127,144]. The accuracy of CTE for the detection of strictures in CD ranges from 78.7% to 83% [145,146]. However, its use is limited by radiation exposure, mainly in the pediatric population [121,126], and previous studies suggest that CTE findings do not correlate with intestinal fibrosis [147]. Meng et al. considered later that this alleged lack of correlation could be related to the focus of the study on the diseased bowel, with reduced attention to the potential value of mesenteric abnormalities on CTE, which, when integrated into a nomogram, could differentiate between non-mild and moderate-to-severe fibrosis in CD patients [148].

In this context of risks and doubts, research was directed to safer and more effective approaches within CT. One of the most valuable explored strategies was the reduction in radiation dose, resorting to high-standard dual-source or ultra-high-pitch CT scanners and iterative reconstruction systems [144,149,150,151,152]. These strategies maintain or improve the quality of CT images and signal-to-noise ratio with lower doses of radiation.

##### Positron Emission Tomography

Even though positron emission tomography (PET) is not considered in current IBD guidelines [153], its ability to detect inflammation in IBD and to add functional data to the structural abnormalities found with MR and CT has supported the research of hybrid techniques, such as PET/MR and PET/CT, in the setting of IBD [136,154,155]. Although promising, their performances in detecting and quantifying fibrosis are globally modest when compared with the results from their single counterparts, especially MR.

Furthermore, high costs and radiation exposure may hinder the wide applicability of these techniques.

##### Ultrasonography

Based on the ability of US elastography to measure tissue stiffness, this technique has been used to evaluate fibrosis in IBD [93,95,113,120]. Several variations of this technique have been studied in the following settings: ultrasound strain elastography (USE) [156,157,158,159,160,161]; shear wave imaging (SWI) [162,163,164]; and contrast-enhanced ultrasonography (CEUS) [160,161,165,166], which have been showing different performances across studies in small cohorts. USE is a non-invasive, innovative technique, in which quantitative measurements of strain ratio can be obtained through the ratio between the strain of a reference region and the strain of the pathologic region, with values above 1 indicating higher stiffness [167]. In the context of IBD, USE showed promising results in animal models [156] and has proven to be able to differentiate between normal and strictured bowel segments [156]. Importantly, it showed to correlate with the severity of fibrosis [157,158,159]. A recent systematic review showed that, in comparison to histopathological assessment, USE showed moderate-to-good accuracy in detecting histological fibrosis [95]. However, based on the available data, the authors considered that USE could not replace the tissue specimen yet, and its applicability must be validated in randomized clinical trials with proper design.

SWI is based on the measurement of shear wave vibration upon application of a pulse wave to tissues by means of an ultrasound probe. The applicability of SWI to fibrosis measurements relies on the fact that transmission of shear waves is faster in stiffer tissues than in softer ones [120]. In CD, SWI was able to distinguish between inflammation from fibrosis in a rodent model [168] but also in human bowel resected from CD patients immediately after surgery [163] and to discriminate between distinct levels of fibrosis in a pilot study with 35 CD patients (*p* = 0.002) [162]. Furthermore, in a comparative study, including three ultrasound techniques, SWI proved to be superior to USE and acoustic radiation force impulse (ARFI) in evaluating and differentiating intestinal stenosis in CD [169]. However, these features could not be confirmed in a pediatric CD cohort [170], and SWI showed no correlation with fibrosis scores in a population of 105 ileal CD patients [164].

Three studies based on the combination of CEUS and USE [160,161] or SWI [171] in the setting of CD showed that combined techniques present an increased ability to differentiate inflammation from fibrosis. The utility of CEUS for fibrosis assessment had been suggested before in a quantitative study in CD patients in which fibrosis seemed to be associated with reduced blood volume and blood flow [172]. However, CEUS did not show similar performance in a 2018 study with 25 CD patients who were evaluated before elective surgery [166]. Despite these conflicting results and the need to recur to an intravenous contrast agent, CEUS seems to add diagnosis value to other imaging techniques. Still, more studies are warranted to explore its full potential.

Even though only a few studies have resorted to Doppler-ultrasonography for the assessment of IBD patients, this technique also deserves mention in this section. In 2013, a study designed to assess the accuracy of US parameters for the evaluation of mural inflammation in CD revealed a significantly negative association between color Doppler grade and fibrosis score (r = −0.584, *p* = 0.001) [165]. Later, Sasaki and colleagues demonstrated that color Doppler was able to predict tissue inflammation and fibrosis in small-intestinal CD lesions (*p* < 0.05) [173].

At this point, CSI techniques have proven to be valuable tools in the attempt to measure fibrosis non-invasively, and ongoing research will be pivotal to defining validated measurement protocols with high accuracy and specificity while guaranteeing minimal risks for patients. In fact, from all the available data and clinical evidence, even though none of the available methodologies is capable of defining the fibrotic component of a CD stricture accurately, MRE-based modalities have proven to be the more advanced for the non-invasive assessment of severe fibrosis in stricturing CD, followed by US-based techniques.

Despite the uncertainties regarding future advances in this field, it is undebatable that the evaluation and diagnosis of fibrosis in IBD requires a multidisciplinary approach involving gastroenterologists, radiologists, pathologists, surgeons, and nurses, among others. Overall, patients benefit from regular monitoring with biomarkers and imaging techniques and from deep clinical discussions in a multidisciplinary setting. In addition, the development of effective referral processes, improved access, and departmental guidelines/pathways with the identification of quantifiable quality indicators creates conditions to provide patients with the best possible diagnosis, treatment, and follow-up [174].

## 4. What Is the Future Holding for Fibrosis?

### 4.1. Radiomics

In a 2020 commentary, Lin and colleagues discussed the concept of computer-assisted image analysis in the context of IBD and suggested radiomics as a tool to transform qualitative fibrosis evaluations on quantitative data [114]. The path to this discussion was opened by a study on the applicability of semi-automated analysis to the measurement of bowel structural damage, with evidence of high consistency with measurements performed by experienced radiologists [117]. In line with data from other diseases, at this point, a few studies support that radiomics of MRE and CTE, consisting of the extraction of high-dimensional data from CSI images, are viewed as a potential source of valuable data for the assessment of IBD fibrosis [115,116,117]. In 2021, Li and colleagues developed a novel CTE-radiomic model for the characterization of intestinal fibrosis in CD, which distinguished the histological non-mild from moderate–severe fibrosis with an AUC of 0.888 on the training cohort and AUCs between 0.724 and 0.816 (95% CI) in the three test cohorts. Moreover, the model performed better than visual interpretations by two experienced radiologists (*p* < 0.001) [116]. The potential of radiomics in this setting was also evidenced in a recent study that reported the integration of CTE on a deep-learning model based on a 3D deep convolutional neural network with 10-fold cross-validation. This model also presented higher accuracy for the assessment of fibrosis severity than CTE evaluation by two radiologists, with the advantage of having a shorter processing time [115]. Despite the robustness of these data, in a letter to the Editor of Gastroenterology, Zhang considered that the validation of these results depends on the development of reliable radiomic biomarkers and criteria to evaluate the design and report of radiomic studies in prospective cohorts [175]. Very recently, the STAR consortium presented the results of a machine-reader evaluation of severe inflammation and fibrosis in CD strictures through quantitative radiomic features and expert radiologist scoring on CTE [176]. Based on the evidenced association of two distinct sets of radiomic features for severe inflammation and fibrosis (*p* < 0.01), the authors considered that the combination of quantitative radiomics with radiological visual assessment might favor more personalized treatments by providing more accurate phenotyping of CD strictures. In the validation study, however, while confirming the value of radiomics in the identification of fibrosis but not inflammation in stricturing CD, the same group did not find advantages in combining radiomic features with the radiologist’s visual assessment [177]. Hence, it seems very likely that radiomics and AI will set the path for the future in the scope of fibrosis assessment by the CSI techniques in CD. However, it is still not clear whether AI alone (without concomitant human intervention) will be able to accomplish this purpose.

### 4.2. Others

Even though imaging techniques have been the focus of most of the developed research regarding fibrosis assessment, studies have diverted to other approaches, such as biochemical and genetic markers, including proteomics, genomics, metabolomics, and transcriptomics. The efforts are supported by the assumption that fibrosis biomarkers would provide useful data for risk stratification and treatment optimization of IBD patients. However, the available evidence includes conflicting data and is focused on markers with low diagnostic and prognostic value. Still, considering that past and ongoing research has provided promising data, an update of the most relevant candidate biomarkers seems appropriate in the context of this revision.

In 2014, the fourth scientific workshop of the European Crohn’s and Colitis Organization (ECCO) focused on understanding basic mechanisms and markers of intestinal fibrosis and considered that, as none of the available biomarkers were able to accurately assess fibrosis, research for novel targets should proceed, as it is pivotal for the development of novel therapeutic options for intestinal fibrosis [119].

In 2012, based on previous findings in the scope of renal disease, Chen and coworkers explored the role of miR-200a and miR-200b in intestinal fibrosis in a colorectal adenocarcinoma epithelial cell line [178]. The results showed that miR-200b was overexpressed in the serum of the fibrosis group and could have diagnostic and therapeutic applications for CD patients with fibrosis. This study leveraged further research on this topic, including a revision on the emerging role of micro-RNAs (miRNAs) in IBD [179], exploring their involvement in the pathways of inflammation and fibrosis in IBD. At that point, the authors considered that miR-200 [178,180] and miR-29b [181] seemed to deserve further research due to evidence of their potential as IBD biomarkers. The importance of these two miRNA families was addressed in two 2015 and 2023 reviews [182,183]. In both, it is stated that the research performed so far still warrants confirmation in more robust studies due to small sample sizes, lack of control of patients’ heterogeneity, and absence of a standard protocol to assess miRNAs.

Other studied candidate biomarkers include serum and plasma proteins, such as collagen [184], ECM [185], pentraxin-2 [186], serum glycoproteins [187], enzymes, such as metalloproteinases [188], antimicrobial antibodies [189,190], and serum growth factors, such as YKL-40 [191,192] and gene variants [193]. Similar to miRNAs, data on these topics are conflicting and do not support their utility in the scope of fibrosis measurement.

In conclusion, considering the vast evidence on this topic, it seems that the efforts on the discovery of novel biomarkers to assess fibrosis would be more consequent through the development of more robust clinical trials based on solid and validated endpoints.

### 4.3. Anti-Fibrotic Therapy

The discussion of fibrosis in the context of IBD can only be completed by addressing the current therapeutic challenges and perspectives toward fibrosis. In the scope of CD, ECCO recommends endoscopic balloon dilatation (EBD) or surgery for patients with short strictures (<5 cm), and strictureplasty for the resection of long segments of the bowel; strictureplasty of the colon is not recommended [194]. Regarding EBD, the PRODILAT study—an RCT with CD patients with the obstructive disease and predominantly fibrotic strictures of less than 10 cm—showed that 80% of the patients approached with this technique were free of a new therapeutic intervention at 1 year; compared with fully covered self-expandable metal stents, EBD proved to be more effective for CD strictures [195].

So far, past and ongoing research did not generate evidence to support the approval of any anti-fibrotic agent. Considering that the fibrosis process is similar in IBD and in systemic and pulmonary fibrosis, several drugs are under investigation as anti-fibrotic agents, in a pre-clinical setting resorting mainly to UC animal models, with promising results in the TGF-β [196,197,198,199,200,201,202,203,204], TNF [205], IL-36 [206], rho-kinase [207], peroxisome-proliferator activated receptor (PPAR) [208], HMG-CoA reductase [209] pathways, among others (Table 2) [5,210]. Table 2 includes the most promising targets and molecules and is not an exhaustive description of all the ongoing research in this field. Regarding phase 2 studies, spesolimab proved to be well tolerated with an adverse event rate similar to placebo (without meeting efficacy criteria) [206], and PF-06480605 demonstrated an acceptable safety profile with concomitant endoscopic improvement (week 14) in patients with moderate to severe UC [205].

Several molecules are now awaiting clinical trials in humans, and in the near future, new therapeutic agents may be approved. Further improvements in this field have been hindered by the reduced research in CD models and by the lack of research standards. The intensive work of the STAR consortium regarding the standardization of the conditions to measure response to anti-fibrotic agents will be determinant for the success of these processes.

## 5. Conclusions

Intestinal fibrosis is a serious complication of IBD with relevant clinical implications that determine treatment selection, prognosis, and quality of life. Currently, available data support the concomitant influence of inflammation-dependent and -independent mechanisms on the induction and progression of fibrosis.

In this review, we highlighted the importance of the development of accurate non-invasive methodologies for the assessment of fibrosis and discussed their strengths, limitations, and future perspectives. In the setting of CSI, MRE advanced modalities seem to be the most robust techniques to measure fibrosis. However, the subjectivity of the visual analysis and interpretation of the images has been hindering the endorsement of CSI in this field. In fact, from the available recent data on radiomics, we believe that imaging techniques will only reach their full potential in terms of accuracy through the combination with artificial intelligence systems. Data in the setting of biomarkers research lack consistency and warrant deeper and more structured trials.

At the end of the day, surgical pathology remains the definitive modality for diagnosing and quantifying intestinal fibrosis in IBD, with the unavoidable disadvantages of being invasive and limiting this study of intestinal damage to “end-of-stage” disease.

## Figures and Tables

**Figure 1 diagnostics-13-02188-f001:**
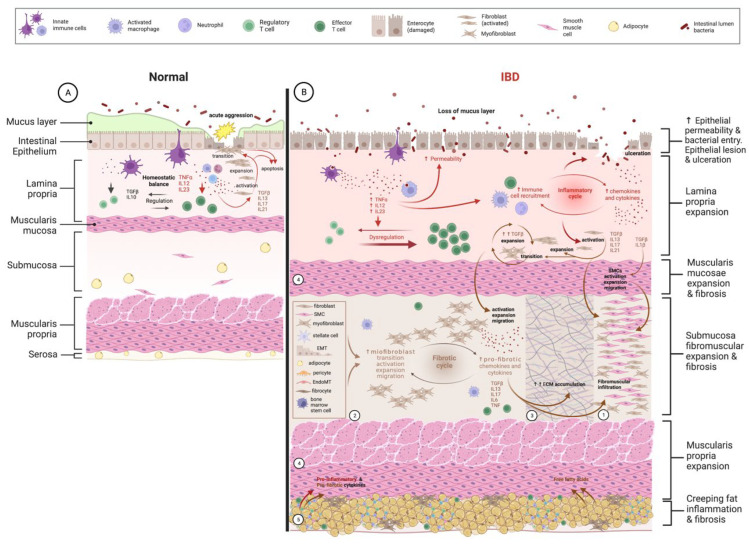
Inflammation-dependent fibrogenesis. (**A**) Left panel. Schematic representation of homeostatic balance between innate and adaptative immune cells in the intestinal lamina propria. Acute aggression to the intestinal epithelium (yellow star) leads to physiological inflammation in view of removing aggression and allowing tissue repair through activation and expansion of local fibroblasts. Part of these will undergo transition to active myofibroblasts, which finalize the restoration of the ECM. When healing is complete, both fibroblasts and myofibroblasts suffer apoptosis. (**B**) Right panel. Schematic representation of dysregulated chronic inflammation occurring in intestinal lamina propria due (among other causes) to increased permeability of the intestinal epithelium, allowing penetration of microbiota perpetuating inflammatory cascades (both cellular and humoral), which also cause local tissue injury (ulceration) with further increase in microbiota access and inflammation on the lamina propria. Chronic inflammation will eventually activate local fibroblasts in the lamina propria, which will expand and migrate to other locations of the intestinal wall, namely, the submucosa. 1. Fibromuscular expansion of the submucosa—due to massive infiltration of activated and expanded fibroblasts, smooth muscle cells (SMCs), and myofibroblasts. 2. Sources of recruitment of activated myofibroblasts driven by intense production of pro-fibrotic mediators by activated myofibroblasts in a vicious-cycle way. 3. Activation of fibroblasts, SMCs, and mostly myofibroblasts leads to chronic and intense production and accumulation of ECM components, mostly on the submucosa, but that may transverse the whole intestinal wall. 4. Activation and expansion of SMCs lead to the thickening of all muscular layers, being disproportional on the muscularis mucosae where fibrosis splaying is usually more common. 5. Creeping fat has recently been demonstrated as a source of both pro-inflammatory and pro-fibrotic mediators, including free fatty acids, which will target both locally, leading to inflammation and fibrosis through creeping fat and on adjacent layers of the intestinal wall. ECM: extracellular matrix; EMT: epithelial-to-mesenchymal transition; endoMT: endothelial-to-mesenchymal transition; IL: interleukin; SMCs: smooth muscle cells; TGFβ: transforming growth factor β.

**Figure 2 diagnostics-13-02188-f002:**
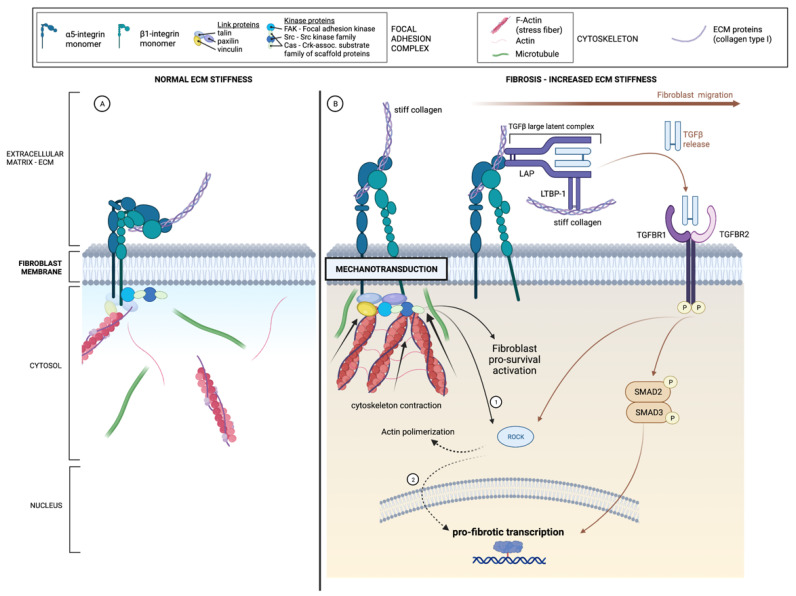
Inflammation-independent fibrogenesis. Schematic representation of mechanotransduction of fibroblasts and latent-TGFβ liberation in response to ECM stiffness. (**A**) Left panel. “Resting state” of focal adhesion complex in normal ECM stiffness. (**B**) Right panel. Mechanotransduction of fibroblasts and latent-TGFβ liberation in response to ECM stiffness. 1. ROCK activation through PI3K and AKT serine/threonine kinase (phosphorylation). 2. ROCK promotes G-actin binding to myocardin-related transcription factor (MRTF), which then migrates to the nucleus. LAP: latency-associated peptide; LTBP-1: latent transforming growth factor β-binding protein; P: phosphorylation; PI3K: phosphatidylinositol 3-kinase ROCK: rho kinase; SMAD: suppressor of mothers against decapentaplegic; TGFβ: transforming growth factor β; TGFBR: TGFβ receptor.

**Table 1 diagnostics-13-02188-t001:** Comparison of the available cross-sectional imaging techniques for the assessment of fibrosis in CD.

Cross Sectional Imaging	Features	Limitations	Future Perspectives
MRE	No radiation High contrast resolution Possibility of performing fluoro-magnetic resonance Can be combined with perianal imaging High accuracy for severe fibrosis identification	Time consuming Intravenous and oral contrast agents Longer scanning time than CTE Less robust than CTE Lower patient compliance than CTE Availability	Validation in more robust clinical trials Combination with radiomics
DWI-MR	Short-time Possible with standard MR scanners No intravenous contrast Qualitative and quantitative analysis High accuracy for inflammation and penetrating complications in IBD High accuracy for severe fibrosis identification	Lack of anatomic details Low reproducibility of ADC Availability	Promising results to be confirmed in more robust clinical trials
DKI-MR	More physiologic imaging No intravenous contrast High accuracy for inflammation Correlation with different fibrosis grades	Few data	Promising results to be confirmed in more robust clinical trials
MT-MR	No intravenous contrast agent Correlation with different fibrosis grades Higher accuracy for fibrosis than MRE with or without DWI	Few data	Promising results to be confirmed in more robust clinical trials
CTE	Accessible Fast Robust Better spatial resolution than MRE	Radiation	Combination with radiomics Reduction in the radiation dose with high-standard dual-source or ultra-high-pitch CT scanners and iterative reconstruction systems
PET/CTE PET/MRE	In combination with CTE or MRE adds functional data	Radiation (labeled marker; CTE) High cost Limited availability Lack of anatomic details	The disadvantages and lack of advantages when compared to CTE and MRE may hinder further developments
USE US-SWI	Real-time visualization of tissue stiffness	Operator dependent Not easy to interpret More difficult to compare current examination with previous studies Heterogeneous data	Promising results to be confirmed in more robust clinical trials
CEUS	Severe fibrosis identification when associated to elastography techniques	Operator dependent Not easy to interpret More difficult to compare current examination with previous studies Heterogeneous data	Promising results to be confirmed in more robust clinical trials

CTE: computed tomography enterography; CEUS: contrast-enhanced ultrasonography; DKI-MR: diffusion kurtosis imaging–magnetic resonance; DWI-MR: diffusion-weighted imaging–magnetic resonance; MRE: magnetic resonance enterography; MT-MR: magnetization transfer–magnetic resonance; PET/CTE: positron emission tomography/CTE; PET/MRE: positron emission tomography/MRE; USE: ultrasound strain elastography; US-SWI: ultrasound–shear wave imaging.

**Table 2 diagnostics-13-02188-t002:** Potential anti-fibrotic agents under research.

Agent	Pathway	Model	Research Status	Reference
Pirfenidone	TGFβ	Human cells	Pre-clinical	[196,197,198,199,200,201]
		Murine models		
		Mice		
Tranilast	TGFβ	Rats	Pre-clinical	[202,203]
		Rat models		
		Patients with CD		
EW-7197	TGFβ	Murine model	Pre-clinical	[204]
PF-06480605	TNF	Patients with UC	Phase 2	[205]
Spesolimab	IL-36	Patients with UC	Phase 2	[206]
AMA0825	Rho-kinase inhibitor	Mice models	Pre-clinical	[207]
		Cells		
		CD biopsies		
GED-0507-34	PPARγa agonist	Mice	Pre-clinical	[208]
Statins	HMG-CoA reductase inhibitors	Human intestinal fibroblasts	Pre-clinical	[209]

CD: Chron’s disease; IL-36: interleukin 36; HMG-CoA: 3-hydroxy-3-methylglutaryl-CoA; PPARγa: peroxisome proliferator-activated receptor-γ; TGFβ: transforming growth factor β; TNF: tumor necrosis factor; UC: ulcerative colitis.

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
