# Peer review of "How to Evaluate Fibrosis in IBD?"

_diagnostics, 2023, doi:10.3390/diagnostics13132188_

Round 1
Reviewer 1 Report
Journal of Diagnostics (MDPI)
Dear EIC, Prof. Dr. Andreas Kjaer
Dear AE,
This is my review report for revised version of manuscript ID: diagnostics-2444132.
In this study, the authors reviewed the underlying immune mechanisms of fibrosis in UC and IBD diseases. They highlighted the importance of fibrosis and reviewed the clinical checkpoints for a better understanding and evaluate of this process in the target population. Although there are a lot of similar papers, this study reviewed the novel edges of the field. The educational content of this study can be useful. The writing is good, and the manuscript flow met standards. Also, figures are well-designed. However, I think one major issue stands out here.
Comments
The authors should design a separate section for reviewing immunotherapy for UC and IBD. Moreover, they should discuss the current immunotherapeutic tools with fibrosis-related mechanisms of action. Also, this section should be supported via table(s) for accessibility and a fast review for readers.
Author Response
Reply to Reviewer´s Comments
Journal: Diagnostics
Title: How to evaluate fibrosis in IBD?
Manuscript ID: diagnostics-2444132
We would like to thank the reviewer for all the comments and suggestions, which were thoroughly accepted. We have introduced the corresponding corrections and clarifications in the new version of the manuscript. The inserted modifications are underlined.
Reviewer 1
- The authors should design a separate section for reviewing immunotherapy for UC and IBD. Moreover, they should discuss the current immunotherapeutic tools with fibrosis-related mechanisms of action. Also, this section should be supported via table(s) for accessibility and a fast review for readers.
Reply: As requested, we inserted a new section dedicated to the treatment of fibrosis (Anti-fibrotic therapy”, under the umbrella of a bigger section called “What is the future holding for fibrosis” that includes also the subsections “Radiomics” and “Others”.
Now in the text:
“Anti-fibrotic therapy
The discussion of fibrosis in the context of IBD can only be completed by addressing the current therapeutic challenges and perspectives toward fibrosis. In the scope of CD, ECCO recommends endoscopic balloon dilatation (EBD) or surgery for patients with short strictures (< 5 cm), and strictureplasty for the resection of long segments of bowel; strictureplasty of the colon is not recommended [195]. Regarding EBD, the PRODILAT study—a RCT with CD patients with obstructive disease and predominantly fibrotic strictures of less than 10 cm—showed that 80% of the patients approached with this technique were free of a new therapeutic intervention at 1 year; compared with fully covered self-expandable metal stents, EBD proved to be more effective for CD strictures[196].
So far, past and ongoing research did not generate evidence to support the approval of any anti-fibrotic agent. Considering that the fibrosis process is similar in IBD and in systemic and pulmonary fibrosis, several drugs are under investigation as anti-fibrotic agents, in a pre-clinical setting resorting mainly to UC animal models, with promising results in the TGF-β [197–205], TNF [206], IL-36 [207], rho-kinase [208], peroxisome-proliferator activated receptor (PPAR) [209], HMG-CoA reductase [210] pathways, among others (Table 2) [5,211]. Table 2 includes the most promising targets and molecules and is not an exhaustive description of all the ongoing research in this field. Regarding phase 2 studies, spesolimab proved to be well tolerated with an adverse events rate similar to placebo (without meeting efficacy criteria) [207] and PF-06480605 demonstrated an acceptable safety profile with concomitant endoscopic improvement (week 14) in patients with moderate to severe UC [206].
Several molecules are now awaiting clinical trials in humans and, in a near future, new therapeutic agents may be approved. Further improvements in this field have been hindered by the reduced research in CD models and by the lack of research standards. The intensive work of the STAR consortium regarding the standardization of the conditions to measure response to anti-fibrotic agents will be determinant for the success of these processes.”
Table 2. Potential anti-fibrotic agents under research
Agent |
Pathway |
Model |
Research status |
Reference |
Pirfenidone |
TGFβ |
Human cells Murine models Mice |
Pre-clinical |
[197-202] |
Tranilast |
TGFβ |
Rats Rat models Patients with CD |
Pre-clinical |
[203, 204] |
EW-7197 |
TGFβ |
Murine model |
Pre-clinical |
[205] |
PF-06480605
|
TNF |
Patients with UC |
Phase 2 |
[206] |
Spesolimab |
IL-36 |
Patients with UC |
Phase 2 |
[207] |
AMA0825 |
Rho-kinase inhibitor |
Mice models Cells CD biopsies |
Pre-clinical |
[208] |
GED-0507-34 |
PPAR?a agonist |
Mice |
Pre-clinical |
[209] |
Statins |
HMG-CoA reductase inhibitors |
Human intestinal fibroblasts |
Pre-clinical |
[210] |
CD: Chron’s disease; IL-36: interleukin 36; HMG-CoA: 3-hydroxy-3-methylglutaryl-CoA; PPAR?a: peroxisome proliferator-activated receptor-γ; TGFβ: transforming growth factor β; TNF: tumour necrosis factor; UC: ulcerative colitis.
Reviewer 2 Report
In the MS ''How to evaluate fibrosis in IBD?' the authors give a brilliant review of the diagnosis of fibrosis in IBD. If I am not mistaken, reading it did not give me any serious concerns. So I have only minor considerations.
It is important to note that the evaluation and diagnosis of fibrosis in IBD often requires a multidisciplinary approach involving gastroenterologists, radiologists and pathologists. Specific assessment methods may vary depending on individual patient factors and available resources. Is this discussion worth recording?
Recently, some research has shown that UC promotes fibrosis without promoting neuron death, but that the chemical code of myenteric neurons is altered (https://doi.org/10.1016/j.lfs.2023.121642). Why is there no approach to the involvement of the enteric nervous system? Is this discussion worth including?
Figures 1 and 2 and their labels are repeated and low resolution.
Author Response
Reply to Reviewer´s Comments
Journal: Diagnostics
Title: How to evaluate fibrosis in IBD?
Manuscript ID: diagnostics-2444132
We would like to thank the reviewer for all the comments and suggestions, which were thoroughly accepted. We have introduced the corresponding corrections and clarifications in the new version of the manuscript. The inserted modifications are underlined.
Reviewer 2
- It is important to note that the evaluation and diagnosis of fibrosis in IBD often requires a multidisciplinary approach involving gastroenterologists, radiologists and pathologists. Specific assessment methods may vary depending on individual patient factors and available resources. Is this discussion worth recording?
Reply: We fully agree with the opinion of the reviewer and, as such, inserted in the Introduction section, a paragraph regarding the multidisciplinary nature of fibrosis assessment.
Now in the text:
“Despite the uncertainties regarding future advances in this field, it is undebatable that the evaluation and diagnosis of fibrosis in IBD requires a multidisciplinary approach involving gastroenterologists, radiologists, pathologists, surgeons, and nurses, among others. Overall, patients benefit from regular monitoring with biomarkers and imaging techniques and from deep clinical discussions in a multidisciplinary setting. In addition, the development of effective referral processes, improved access, and departmental guidelines/pathways with the identification of quantifiable quality indicators creates conditions to provide patients with the best possible diagnosis, treatment, and follow-up [175].”
- Fiorino, G.; Lytras, T.; Younge, L.; Fidalgo, C.; Coenen, S.; Chaparro, M.; Allocca, M.; Arnott, I.; Bossuyt, P.; Burisch, J.; et al. Quality of Care Standards in Inflammatory Bowel Diseases: A European Crohn’s and Colitis Organisation (ECCO) Position Paper. J Crohns Colitis 2020, doi:10.1093/ecco-jcc/jjaa023.
- Recently, some research has shown that UC promotes fibrosis without promoting neuron death, but that the chemical code of myenteric neurons is altered (https://doi.org/10.1016/j.lfs.2023.121642). Why is there no approach to the involvement of the enteric nervous system? Is this discussion worth including?
Reply: We thank you the reviewer for the highlight. In line with this recent advance, we inserted, in the section “Fibrosis in UC”, a sentence regarding the need for further studies to validate the involvement of the enteric nervous system.
Now in the text:
“Recent research in mice with dextran sulfate sodium (DSS)-induced colitis has shown that, in UC, changes in motility may also be related to neuronal modification. The study highlighted that UC does not promote neuron death but induces changes in the chemical code of myenteric neurons [41]. A better comprehension of these data and the translation of these results depend on studies in human tissue.”
- Figures 1 and 2 and their labels are repeated and low resolution.
Reply: In the sequence of this comment, we confirmed that the figures present adequate resolution; the repeated figures were removed from the text.
Round 2
Reviewer 1 Report
Dear EA,
My comments were corrected and I convinced by the authors feedback.